# Accelerating Effect of *Cucurbita pepo* L. Fruit Extract on Excisional Wound Healing in Depressed Rats Is Mediated through Its Anti-Inflammatory and Antioxidant Effects

**DOI:** 10.3390/nu14163336

**Published:** 2022-08-15

**Authors:** Hailah M. Almohaimeed, Maryam Hassan Al-Zahrani, Mohammed Saad Almuhayawi, Sami Awda Algaidi, Ashwaq H. Batawi, Hasan Ahmed Baz, Zuhair M. Mohammedsaleh, Nhal Ahmed Baz, Fayez M. Saleh, Nasra Ayuob

**Affiliations:** 1Department of Basic Science, College of Medicine, Princess Nourah bint Abdulrahman University, Riyadh 11671, Saudi Arabia; 2Department of Biochemistry, Faculty of Science, King Abdulaziz University, Jeddah 21589, Saudi Arabia; 3Department of Medical Microbiology & Parasitology, Faculty of Medicine, King Abdulaziz University, Jeddah 21589, Saudi Arabia; 4Yousef Abdullatif Jameel Chair of Prophetic Medicine Applications (YAJCPMA), Faculty of Medicine, King Abdulaziz University, Jeddah 21589, Saudi Arabia; 5Department of Anatomy, Taibah University, Medina 42353, Saudi Arabia; 6Department of Biological Science, Faculty of Science, King Abdulaziz University, Jeddah 21589, Saudi Arabia; 7Clinical Toxicology, Faculty of Medicine, Umm Al Qura University/King Abdullah Medical City, Mecca 24211, Saudi Arabia; 8Department of Medical Laboratory Technology, Faculty of Applied Medical Sciences, University of Tabuk, Tabuk 71491, Saudi Arabia; 9Department of Periodontist, King Abdullah Medical City, Mecca 24211, Saudi Arabia; 10Department of Medical Microbiology, Faculty of Medicine, University of Tabuk, Tabuk 71491, Saudi Arabia; 11Department of Medical Histology, Faculty of Medicine, Damietta University, Damietta 34517, Egypt

**Keywords:** depression, healing, pro-inflammatory, anti-inflammatory, antioxidants, cytokines, wound, antidepressant

## Abstract

Background: Chronic stress can hinder wound healing as it suppresses both the cellular and innate immune responses. Objectives: The study aims to assess the effectiveness of the administration of topical and oral *Cucurbita pepo* L. (CP) ethanolic extract in prompting excisional wound healing in rats exposed to chronic stress, and to explain how it works. Materials and methods: Fifty albino rats assigned to five groups (*n* = 10) were utilized in this study. The chronic unpredictable mild stress (CUMS) model was used for 4 weeks to induce depressive-like behavior in rats, and a forced swim test and corticosterone were assessed to confirm its occurrence. During the experiment, an excisional wound was induced in the rats and followed. Oxidant/antioxidants status and pro-inflammatory cytokines levels were measured in the serum and wound area. Gene expression of pro-inflammatory cytokines was also assessed using RT-PCR. Wound closure histopathological changes and immunohistochemical expression of CD68, CD3, and CD4 at the wound area was assessed. Results: The administration of CP, both orally and topically, significantly reduced (*p* < 0.001) the depressive-like behavior and corticosterone and pro-inflammatory cytokines levels, while it significantly up-regulated the antioxidant activity compared to the untreated and topically CP-treated groups. Both topically CP-treated and combined CP-treated groups showed complete re-epithelialization, reduced inflammatory cells infiltration, collagen fibers deposition, and significantly increased CD3, CD4 positive T cells count, with a superior effect in the combined CP-treated groups. Conclusion: *Cucurbita pepo* L., administrated both topically and orally, can enhance the wound healing process in rats with depressive-like behavior mostly through the antioxidant, anti-inflammatory, and antidepressant activities observed in this study.

## 1. Introduction

Depression is considered the number one cause of disability in the world. Medical illness, particularly metabolic and autoimmune diseases, increase the risk of depression [1].

Many observations of wound healing in both animal and human studies have revealed that psychological stress may act as a deterrent to wound healing. The proposed mechanism describes possible physiological pathways through which stress changes the process of wound healing [2]. During surgery, under chronic stressful conditions, debris might be declared slowly with sustained low-grade inflammation. In addition, both cellular and innate immune responses are suppressed in these conditions, with the subsequent delay in the wound healing process [3].

The pro-inflammatory cytokines, including interleukins IL1α, IL 1β, IL-6, and TNF-α, have a crucial role in the process of wound healing. They impact various steps at the wound site, including the proliferation of keratinocyte and fibroblast, synthesis and breakdown of extracellular matrix proteins, fibroblast chemotaxis, and regulation of the immune response [4]. Pro-inflammatory cytokines, e.g., IL-1β, IL-6, TNF-α, and acute phase proteins, were described to be increased in experimental animal models of chronic stress-induced depression [5] as well as in patients with major depressive disorder (MDD) compared to healthy controls [6]. Therefore, the pro-inflammatory cytokines were evaluated in this study in order to explore the mechanism of action of pumpkin as an antidepressant.

Although there are many topical preparations that have wound healing properties on the market, an appropriate drug that provides effective, less toxic, low-cost, rapid and safe effect still lacking [7]. Most of the topical preparations available in the market have antimicrobial activities rather than wound repair effects [8].

The Cucurbitaceae family, including *Cucurbita pepo* L., is considered a rich plant source of antioxidants and vitamins [9]. The different types of pumpkin leaves extract, water, ethanol, hexane, showed a potent antibacterial activity as they inhibit the growth of many bacteria, e.g., Escherichia coli, Klebsiella pneumonia, Staphylococcus aureus, Proteus mirabilis and Pseudomonas aeruginosa, as well as fungi, e.g., Aspergillus fumigatus, Aspergillus niger, and Candida albicans comparable with the known antibiotic Ciprofloxacin and the antifungal drug Kenazole [10]. It was postulated that the compounds that have antimicrobial, anti-inflammatory, and antioxidant activities endorse the wound healing process. Astringent and hemostatic agents inhibit excessive bleeding by providing wound closure, specifically, in the early phases of wound healing [11]. In addition, the antidepressant fluoxetine was reported to inhibit the negative effects of stress on tendon healing [12].

*Cucurbita pepo* L. is one of the plants that possess antimicrobial, anti-inflammatory, and antioxidant activities [13]. These activities could explain its efficacy in enhancing wound healing in different wound models that included excision wound [14,15,16] and burns wound [17,18]. Additionally, pumpkin has previously been described as having antidepressant [19] and anti-fatigue activities [20]. Hence, pumpkin was studied in this experiment to evaluate its efficacy in promoting wound healing in rats with depressive-like behavior. Therefore, we hypothesized that *Cucurbita pepo* L. could enhance excisional wound healing in an animal model of depression induced by exposure to chronic unpredictable mild stress (CUMS) through its anti-inflammatory and antioxidant activities.

## 2. Materials and Methods

### 2.1. Extraction of Cucurbita pepo L. (CP)

Fresh CP fruits were purchased from the local market in Jeddah, Saudi Arabia (voucher specimen: AQJ_123) and their species was verified by a specialist at the Botany department, Faculty of Science, King Abdulaziz University, Jeddah, Saudi Arabia.

Extraction of CP was performed based on the method of Wang et al. [20], as described in previous work [21]. The skin and pulp of the fruit were dried by a lyophilize machine freeze-drier and crushed with a grinding electrical machine. The dried powder (50 g) was mixed with 450 mL ethanol (80%) for 1 day at 37 °C temperature, left in a shaker machine for another day, and then filtered twice. The extract was left in a fume hood to allow extra evaporation of ethanol; then, the extract was dried in a freeze-drier. The extract was stored in a glassy container until the time of use where it was diluted with hot distilled water (2:1) in an ultrasonication bath and administrated orally (100 mg/kg) in 1 mL once daily, in the morning, for two weeks [21].

### 2.2. Preparation of Cucurbita pepo L. for Topical Application

The ethanolic extract of CP was formulated in Vaseline at a proportion of 2% (*w*/*w*) in order to prepare a simple ointment using a ceramic pestle and mortar, then it was stored in a sterile container. It was applied topically on the skin wound once daily for two weeks [17].

### 2.3. Induction of Skin Excisional Wound

Before performing the excisional wound in the rat, induction of anesthesia was made through 4% isoflurane (SEDICO Pharmaceuticals Company, Cairo, Egypt) in 100% oxygen. Under aseptic conditions, the dorsal fur of the rat was shaved, and then a full thickness wound of 1.5 cm in diameter and 0.2 cm in depth was made using a sterile surgical blade and scissors. The entire wound was left open and treated topically with the drug formulations after 24 h. Hemostasis was carried out by blotting the wound with a cotton swab soaked in normal saline.

The morphological changes in the wound area were observed and monitored by the mobile camera every other day. The wound area was measured by a clear sterile ruler placed next to the lesion to serve as a reference for measurements. The wound closure process was followed by taking a photograph every 2 days. The percentage of wound closure, taking the initial size of the wound as 100%, was calculated by using the following equation according to the method of George et al. [22].
$$\frac{\%\;\mathrm{of}\;\mathrm{wound}\;\mathrm{closure}=\mathrm{initial}\;\mathrm{wound}\;\mathrm{size}\;-\;\mathrm{specific}\;\mathrm{day}\;\mathrm{wound}\;\mathrm{size}\;\times \;100}{\mathrm{Initial}\;\mathrm{wound}\;\mathrm{size}}$$

### 2.4. Experimental Groups and Dosage

Fifty male albino rats with a weight ranging from 150 to 200 g and an age range of 2 to 3 months were purchased from the King Fahd Medical Research Center (KFMRC) and utilized in this study. It was reported that the response to stress differs between genders [23]. Therefore, in this study, only male rats were used. Ethical approval of the study protocol was obtained from the biomedical research ethics committee at King Abdulaziz University, Jeddah, Saudi Arabia (reference number 45–20). Rats were kept in cages in an air-conditioned animal house at 22 ± 1 °C, with free access to the standard rat chow and water ad libitum for one week to acclimatize to the laboratory conditions. They were divided into a negative control group (*n* = 10) that was not exposed to stress and their dorsal skin was shaved, then cleaned daily with saline solution. The forty rats were exposed to the technique of chronic unpredictable mild stress (CUMS), which included exposure to different types of stressors at different times during the day to prevent the habituation of rats to the stressors as was previously mentioned [24].

The forty CUMS-exposed rats were further divided into 4 groups (*n* = 10) (Figure 1). Rats of the positive control group were exposed to CUMS for 4 weeks then an excisional wound was induced and treated with the vehicle Vaseline ointment alone for 2 weeks, once per day. Rats of the reference group were exposed to CUMS; then an excisional wound was induced and treated with the standard ointment Betadine (5% *w*/*w*, The Nile Co. for Pharm. Cairo, Egypt). Rats of the CP-treated group were exposed to CUMS then an excisional wound was induced and locally treated with CP ointment according to the method described by Bahramsoltani et al. [17]. Rats of the combined CP-treated group were exposed to CUMS then an excisional wound was induced and locally with CP ointment plus oral CP extract (100 mg/kg) through gavage, as described by Wang et al. [20]. Formulations were topically applied to cover the wound surface every 12 h for 14 days.

### 2.5. Assessment of CUMS-Induced Behavioral Changes

Depressive-like status was evident in the rats exposed to CUMS using the forced swim test (FST), at the end of the experiment, as previously described by Yankelevitch-Yahav et al. [25]. While being left to swim in a glassy cylindrical container, the rat was observed for 6 min and the total time spent by the rat without mobility during the 6 min was calculated. Immobility was defined as the cessation of limb movement, except for the minor movement necessary to keep the rat afloat.

### 2.6. Assessment of the CUMS-Induced Biochemical Changes in the Serum

At the end of the experiment, the mice were anesthetized using 4% isoflurane, and the blood sample was obtained in the morning from the retrorbital venous plexus into EDTA coated tubes. They were then centrifuged for 10 min, and the collected serum samples were kept at −80 °C.

Serum corticosterone level was assessed during the experiment in order to confirm the occurrence of depressive-like status, using enzyme-linked immunosorbent assay (ELISA) kits (ALPCO Diagnostics, Orangeburg, NY, USA) based on the manufacturers’ instructions.

Serum levels of tumor necrosis factor-α (TNF-α) and IL-6 (quantakin R & D system, USA Kit) were measured using ELISA.

The level of malondialdehyde, Superoxide dismutase (SOD), and catalase (CAT) were assessed using assay kits (Biodiagnostic, Egypt) based on the method described by Gamal et al. [26] and Packer et al. [27].

Glutathione peroxidase (GPX) was assessed using assay kits (Randox Labs, Crumlin, UK) according to the method described by Gamal et al. [26].

### 2.7. Assessment of CUMS-Induced Biochemical Changes in the Skin

Samples of the edges of the wound were excised and kept at −80 °C for measurement of protein. The samples were homogenized and centrifuged for 10 min at 5000× *g*. The supernatant was utilized to assess the levels of TNF-α, IL-6, SOD activity, CAT, and GPX in the skin using the ELISA assay Kit (Biodiagnostic; Giza, Egypt) as was described above.

### 2.8. Quantitative Real-Time PCR (qRT-PCR)

One hundred milligrams of formalin-fixed paraffin-embedded sections was utilized for RNA extraction, de-paraffinized in 1 mL of xylene, and incubated at 56 °C for 15 min, centrifuged for 10 min at 13,000× *g*. The pellets were washed twice, after discarding the supernatant, with 1 mL 100% ethanol, centrifuged, and mixed with 1 mL Trizol [28]. Trizol (Invitrogen Life Technologies, Carlsbad, CA, USA) was used to extract the total RNA according to the supplier instruction and was described in a previous study [29]. Amplification of the cDNAs was performed using PCR Master Mix (Bioneer’s AccuPower® PCRm Oakland, CA 94607 Bioneer). The primers used were prepared by Metabion International (Semmelweisser, 3 Planegg, 82152 Germany) and included; rat-TNF-α: (5′-CCCTGGTACTAACTCCCAGAAA-3′; 5′-TGTATGAGAGGGACGGAACC-3′), rat-IL-6 (5′-CTGCAAGAGACTTCCATCCAG-3′; 5′-AGTGGTATAGACAGGTCTGTTGG-3′), rat-iNOS (5′-CACCACCCTCCTTGTTCAAC-3′; 5′-CAATCCACAACTCGCTCCAA-3′), rat-COX2 (5′-TGCGATGCTCTTCCGAGCTGTGCT-3′; 5′-TCAGGAAGTTCCTTATTTCCTTTC-3′) and rat-GAPDH (5′-CAACTCCCTCAAGATTGTCAGCAA-3′; 5-′GGCATGGACTGTGGTCATGA-3′). The PCR reactions were kept track of by determining the strength of the fluorescence brought on by SYBR Green Dye intercalation to the double-stranded DNA (dsDNA), and melting curve evaluation was carried out to verify the specificity of the products. The level of mRNA was presented as a ratio or percent to that of the corresponding GAPDH

### 2.9. Histopathological Examination of the Wound Area

After completing 14 days of treatment, all rats were decapitated under anesthesia. The edges of the wound were dissected out (about 2 × 2 mm) and immersed in 10% neutral buffered formalin, and processed into paraffin blocks that were sectioned at 4 µ and stained with haematoxylin and eosin (H&E), Masson trichrome for visualization of collagen fibers.

A set of the slides were immunohistochemically stained using the streptavidin–biotin–peroxidase technique. Anti CD3, Anti CD4, anti-CD68 antibodies (Biocare Medical, Pacheco, USA at dilution of 1:100) were used. The primary one was omitted during staining of some slides while the secondary antibody IgG was added to act as a negative control slide. Hematoxylin was used for counterstaining. Brown cytoplasmic staining indicated a positive reaction.

Olympus Microscope BX-51 supplied with a digital camera was used for photographing the histopathological changes in the wound healing process by a histologist. Quantification of immunoexpression of the antibodies was performed by Pro Plus image analysis software (Media cybernetics, Rockville United States of America (USA). The number of immunopositive cells was counted in 30 fields (at 400× magnification) then the mean was calculated for each rat [30].

### 2.10. Statistical Analysis

The Statistical Package for the Social Sciences (SPSS) version 16 was used, in this study, to analyze the quantitative study variables, including the behavioral, biochemical, and the semi-quantitative variables included the immunohistochemical assessment. Normality of the data was assessed, then analysis of variance was used to compare the parametric data followed by a Bonforoni post hoc test. The Mann–Whitney *t*-test was used to compare the nonparametric data. Statistical significance was considered at *p* < 0.05.

## 3. Results

### 3.1. The Effect of Combined CP Administration on the CUMS-Induced Behavioral Changes

In this study, statistical analysis revealed a significant increase (*p* < 0.001) in immobility time of the FST in the positive control group following exposure to CUMS compared to the negative control group. The administration of neither reference cream nor CP cream did not affect the immobility time, while oral administration of CP in the combined CP-treated group significantly reduced (*p* < 0.001) the immobility time compared to the positive control group (Table 1).

### 3.2. The Effect of Combined CP Administration on the Wound Size and Enhances Wound Closure

The macroscopic picture of the progress of the wound healing process in the study groups is presented in Figure 2.

In order to assess the wound healing activity, the wound area was assessed every two days, and the values on days 4 and 14 are shown in (Table 1). The percentage of wound closure at the end of the experiment was calculated and compared for significance. It was noticed that the percentage of wound closure in the positive control group was only 47%. It showed a significant increase (*p* < 0.001) in all the treated groups compared with the positive control group with the highest significance in the combined CP-treated group. There was a significantly higher wound closure percentage in the combined CP-treated group compared with the CP-treated group (*p* = 0.01) (Table 1).

### 3.3. The Effect of Combined CP Administration on the CUMS-Induced Effect on Corticosterone

In this study, the serum level of corticosterone was significantly increased (*p* < 0.001) in the positive control group compared to the negative control as a result of exposure to CUMS. Administration of CP cream combined with oral CP significantly reduced (*p* < 0.001) corticosterone levels compared to the positive control and the CP-treated group with ointment alone. The corticosterone level showed no significant change in the reference group and CP-treated compared to the positive control group.

### 3.4. The Effect of Combined CP Administration on the Pro-Inflammatory Cytokines

In this study, exposure to CUMS and then excisional wound resulted in a significant increase in the level of pro-inflammatory cytokines IL-6 (*p* < 0.001) and TNF-α (*p* < 0.001) in both serum and skin compared to the negative control group. The level of IL-6 and TNF-α was significantly reduced in the combined CP-treated (*p* < 0.001) group in the serum (*p* < 0.001) and skin (*p* < 0.001) compared with the control group. The serum and skin levels of IL-6 and TNF-α did not show a significant change in the reference group compared to the positive control one (Table 1).

In alignment with these results, the level of mRNA of both IL-6 (*p* < 0.001) and TNF-α (*p* < 0.001) was significantly up-regulated in the skin of the wound area of the positive control group compared to the negative control. The level of mRNA of IL-6 (*p* < 0.001 and *p* = 0.01) and TNF-α (*p* < 0.001) and was significantly reduced in both CP-treated and combined CP-treated (groups compared to the positive control group. There was no significant difference in mRNA levels of IL-6 and TNF-α of the reference group (Figure 3).

In this study, the positive control group showed a significant increase in skin level of iNOS (*p* < 0.001), a common indicator of wound hypoxia, and COX-2 (*p* < 0.001) compared to the negative control group. On the other hand, iNOS, and COX-2 levels were significantly reduced in CP-treated (*p* = 0.004, *p* = 0.01) and combined CP-treated (*p* < 0.001) groups in, respectively, with a significant difference (*p* = 0.04, *p* = 0.03) between the two groups (Table 1).

These findings were confirmed with that of the gene expression study as mRNA of iNOS (*p* < 0.001) and COX-2 (*p* < 0.001) was significantly up-regulated in the skin of the wound area of the positive control group compared to the negative control. mRNA of iNOS and COX-2 showed a significant reduction in both CP-treated (*p* = 0.01) and combined CP-treated (*p* = 0.01, *p* < 0.001) groups compared to the positive control group. There was no significant difference in the mRNA level of iNOS and COX-2 of the reference group (Figure 3).

### 3.5. The Effect of Combined CP Administration on the CUMS-Induced Oxidative Stress

In this study, MDA was significantly increased (*p* = 0.001, *p* = 0.01) in the positive control group, while SOD (*p* < 0.001), CAT (*p* = 0.001), GPX (*p* = 0.002, *p* < 0.001) significantly reduced in serum and skin, respectively, compared with the negative control (Figure 4).

A significant reduction (*p* = 0.002) in skin MDA level of the CP-treated group was observed, while its serum level did not show significant changes compared to the positive control group. Both skin and serum levels of MDA were significantly reduced in the combined CP-treated group (*p* = 0.001, *p* = 0.003) compared to the positive control group.

Levels of SOD (*p* < 0.001), CAT (*p* < 0.001), GPX (*p* = 0.002) were significantly increased in the skin of CP-treated, while their serum levels did not show a significant difference in this group compared to the positive control one. Combined CP-treated group showed a significant increase in both skin and serum levels of SOD (*p* < 0.001), CAT (*p* < 0.001, *p* = 0.001), GPX (*p* < 0.001), with a significant difference (*p* = 0.002, *p* = 0.04, *p* = 0.01) between the CP-treated group, respectively (Figure 4).

### 3.6. The Effect of Combined CP Administration on CUMS-Induced Negative Impact on the Wound Healing Process and Immunity

In this study, histopathological examination of the wound area after 14 days of wounding revealed that the positive control and reference groups showed incomplete re-epithelization, many neutrophils, inflammatory cells, few fibroblasts, and irregular and sparse collagen fibers. On the other hand, the wound areas of both CP-treated and combined CP-treated groups showed a complete newly formed epithelial layer, dominated by fibroblasts and few inflammatory cells. Masson stain revealed the presence of adequate, well-arranged, and compact collagen fibers (Figure 5). There was a significant increase (*p* = 0.01, *p* < 0.001) in the area percent of Masson trichrome-stained collagen fibers of CP-treated and combined CP-treated groups, respectively, compared with the positive control (Table 1).

In order to confirm the ability of CP to alleviate the stress-induced disturbance in immune cells during wound inflammation, staining of CD3, CD4, the immunohistochemical markers of T cells, and CD68, the immunohistochemical markers of macrophages were performed in the study groups. It was found that the number of CD68 positive macrophages significantly increased (*p* < 0.001) in the positive control group as a result of exposure to CUMS, while it significantly reduced (*p* = 0.02, *p* < 0.001) in CP-treated as well as combined CP-treated groups compared with the positive control one, respectively. (Table 1).

The number of CD3-, CD4-positive lymphocytes showed a significant reduction (*p* = 0.02, *p* = 0.01) in the positive control group, while it showed a significant increase in CP-treated (*p* = 0.02, *p* < 0.001) and combined CP-treated (*p* < 0.001) groups (Table 1).

## 4. Discussion

The process of healing soft tissue wounds includes hemostasis/inflammation, migration, proliferation, and remodeling [31]. It was postulated that all stages of wound healing are affected by the psychobiologic effects of stress evidenced by cytokine/immune crosstalk dysregulation, hypoxia, dysregulation of cellular mobility, metabolomic kinetics, and matrix metalloprotease activity [3]. This study was performed to assess the role of the combined local and systemic administration of *Cucurbita pepo* in enhancing wound healing in a chronic stress-induced animal model of depression and explain the mechanism of this role.

In this study, after exposure to CUMS, rats showed depressive-like status revealed by increased immobility duration during FST, as well as the marked increase in serum corticosterone that pointed to the hyperactivity of the hypothalamic–pituitary–adrenal (HPA) axis. These findings probably were in agreement with many previous studies [32]. The CUMS model was used in this study as it has become firmly established as an indispensable experimental tool for studying the neurobiological basis of depression with proved validity and reliability [32].

In order to determine if CP could alleviate the stress-induced increase in wound inflammation, inflammatory cytokines IL-6 and TNF-α were assessed in this study. It was noticed that exposure to CUMS and excisional wounds resulted in a significant systemic and local increase in inflammatory cytokines IL-6 and TNF-α. This finding is probably in accordance with that of Zhao et al., 2012 who noted increased levels of inflammatory mediators IL-6 and TNF-α in an animal model of experimental periodontitis exposed to chronic stress conditions [33]. It was reported that expression of IL-1α, IL-1β, IL-6, and TNF-α was shown to be strongly up-regulated during the inflammatory phase of healing. The polymorph nucleated cells (PMN) and macrophages were shown to be the major source of these cytokines. The coordinated expression of these cytokines is likely to be important for normal repair [34]. Stress also was described to increase the levels of monocyte chemotactic protein-1 (MCP-1), an important factor of the inflammatory response in cutaneous lesions [35]. Increased levels of stress hormones were reported to increase the duration of inflammatory responses and compromise the development of its subsequent phases [36].

A significant increase of iNOS in the skin of the wound, which indicates wound hypoxia, was observed in this study at the level of both protein and gene expression. This was previously described in some experimental animal models of chronic stress [37]. The link between iNOS and COX-2 during the process of wound healing was previously described. Romana-Souza, Dos Santos [38] postulated that selective inhibition of COX-2 by celecoxib has been shown to improve wound healing of pressure ulcers by reducing the expression levels of iNOS [38].

In this study, it was observed that the CUMS delayed wound healing based on wound closure percentage. Histopathologically, delayed wound healing was evident by incomplete re-epithelization, with many neutrophils, inflammatory cells, few fibroblasts, and irregular and sparse collagen fibers being observed during the wound healing process after exposure to CUMS. This is in agreement with the findings of some experimental studies—cutaneous punch biopsy wounds in immobilized Siberian hamsters housed alone [39] and rational stress model [37] social stress [40]. In these previous stress models, the negative impact of psychological stress was attributed to reducing collagen deposition [30,36].

This negative impact might be attributed to increased corticosterone, which was also observed in this study. Mercado, Quan [41] reported that stress-induced glucocorticoid hormones in mouse models affect the inflammatory phase of wound repair, suppress the recruitment of inflammatory cells to the wound margin, hinder antibacterial function, and delay healing [41]. Sevilla and Pérez [42] also reported that increased glucocorticoid levels decrease keratinocyte expression of the growth factors and cytokines necessary for re-epithelialization following injury, delaying wound healing and render the area susceptible to infection [42].

The association between stress and depression and “immune suppression”, presented by reduced proliferative responses of immune cells or “immune activation” manifested by the proliferation of immune cells and the increased production of pro-inflammatory cytokines was described to be well supported and potentially clinically relevant observations in depression [43]. The impact of chronic stress-induced depression on the local immunity of the wound was assessed immunohistochemically in this study. We observed an increase in the number of CD68-positive macrophages as well as a reduction in the number of CD3- and CD4-positive T cells at the site of the wound. In support of this, it has been previously reported that psychological stress increases the migration of neutrophil and macrophage and lipid peroxidation in the wound area of mice and reduces the number of CD-3 and CD-4 positive T cells and TNF-α protein levels in the wound area [31,37].

Although depression and anxiety are potential stimuli for cortisol release, they can have adverse immunological effects [44], however, the topic is contentious as depression has been shown to be associated with increased release of pro-inflammatory cytokines, diminished glucocorticoid sensitivity, immune deregulation, and down-regulation of both cellular and humoral immune responses [45]. Decker, Kapila [3] recently reported that chronic stress in the case of depression and post-traumatic stress disorder (PTSD) often presents with suppression of both the cellular and innate immune responses as CD4- and CD8-positive T helper cells [3].

Although wound closure percentage and wound-healing properties of pumpkin have been previously reported in different models of wounds [14,46], none of these studies proved this effect in the presence of chronic stress or depressive-like behavior. In this study, wound closure percentage showed a significant increase in the rats CP-treated with CP, with the superior effect of the combined CP local and oral administration of CP. This might be attributed to the antidepressant effect of pumpkin, reported in the previous study [19,47], as well as the antioxidant and anti-inflammatory effects of pumpkin [48] that are more evident with systemic administration. It has been reported that oleic and linoleic acids are among the compounds that exhibit strong pro-inflammatory activities and play a major role in the recruitment of inflammatory cells at the inflammation site that lead to accelerating the wound healing process. Fortunately, pumpkin possesses a high concentration of these two compounds [49,50].

In this study, a depressive-like status resulted from exposure to CUMS for four weeks was associated with dis-regulation in oxidants/antioxidant status evidenced by increased serum MDA and reduced SOD, CAT, and GPX. Similar changes were reported in the animal models of depression by [51]. Topical administration of CP, in this study, induced a significant increase in the skin levels of enzymatic antioxidants; SOD, CAT, and GPX, as well as a significant reduction in MDA, while the combined oral and topical administration of CP significant increased these antioxidants in both skin and serum. The antioxidant effect of pumpkins was confirmed by its up-regulation of the activities of antioxidant enzymes such as GSH-Px and SOD and reduced MDA [13,17,52]. This explained the superior effect of the combined administration of CP, evident in this study, as it induced a significant systemic antidepressant, antioxidant and anti-inflammatory effect when absorbed by the oral route.

The antioxidant activity of CP is one of the proposed mechanisms that explain its efficacy in wound healing that was observed in this study. The phenolic and flavonoid compounds of pumpkin specifically are among the essential elements in the healing of wounds due to their potential anti-inflammatory and antioxidant effects [11]. In our previous study, the essential fatty acids linolenic and linoleic acids were detected in *Cucurbita pepo* L. [21]. These long-chain polyunsaturated fatty acids participate in the formation of healthy cell membranes and rapid wound healing activity [53]. Adding to that, *Cucurbita pepo* L. has antimicrobial activity that represents another mechanism of enhancing wound healing, as it forms a barrier against microbial contamination [13]. It has been described that the high mucilage content of pumpkin is able to accelerate wound healing as it provides a moist environment and stimulates re-epithelialization, angiogenesis, keratinocyte migration, and induction of hypoxia-inducible factor-1 (HIF-1) [17,54].

In conclusion, this study revealed that the ethanolic extract of *Cucurbita pepo* L. succeeded in enhancing the wound healing process in rats with chronic stress-induced depressive-like behavior with a superior effect when administrated both locally and orally. The significant antioxidant, anti-inflammatory and antidepressant activities of *Cucurbita pepo* L., proved in this study, represent potential mechanisms that could explain this effect. A future study to test this effect in humans is recommended.

## Figures and Tables

**Figure 1 nutrients-14-03336-f001:**
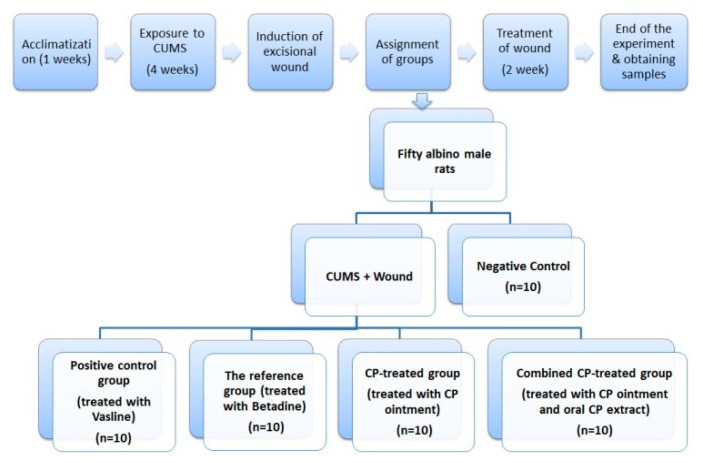
Design and grouping of the experiment. CP: *Cucurbita pepo* L.

**Figure 2 nutrients-14-03336-f002:**
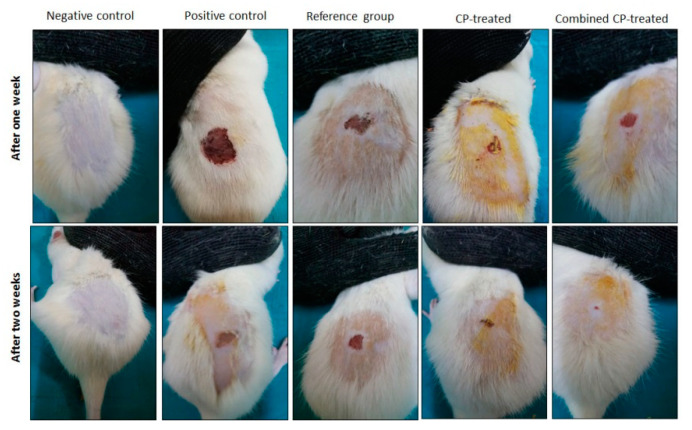
Macroscopic progress of wound healing process in the study groups. CP: *Cucurbita pepo* L.

**Figure 3 nutrients-14-03336-f003:**
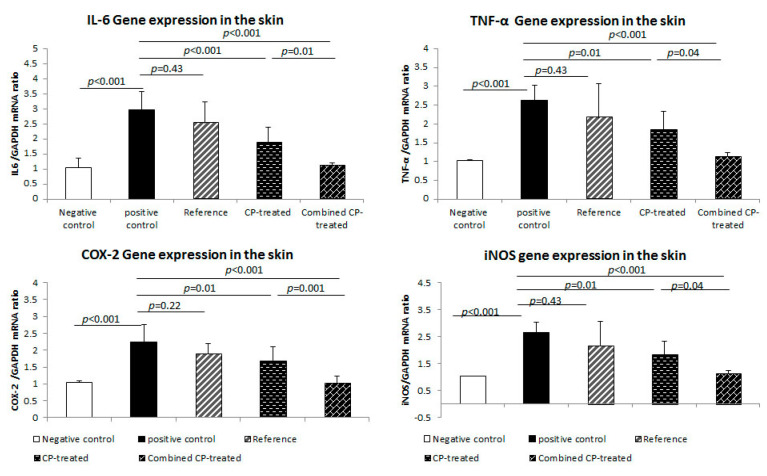
Effect of *Cucurbita pepo* L. (CP) on the gene expression of the inflammatory cytokines of the study groups. The levels of mRNA of IL-6, TNF-α, COX-2, iNOS were assessed in the skin using qRT-PCR. Data are presented as the mean ± SD, *n* = 10. One way ANOVA test was used to compare between groups, followed by the Bonferroni post hoc test.

**Figure 4 nutrients-14-03336-f004:**
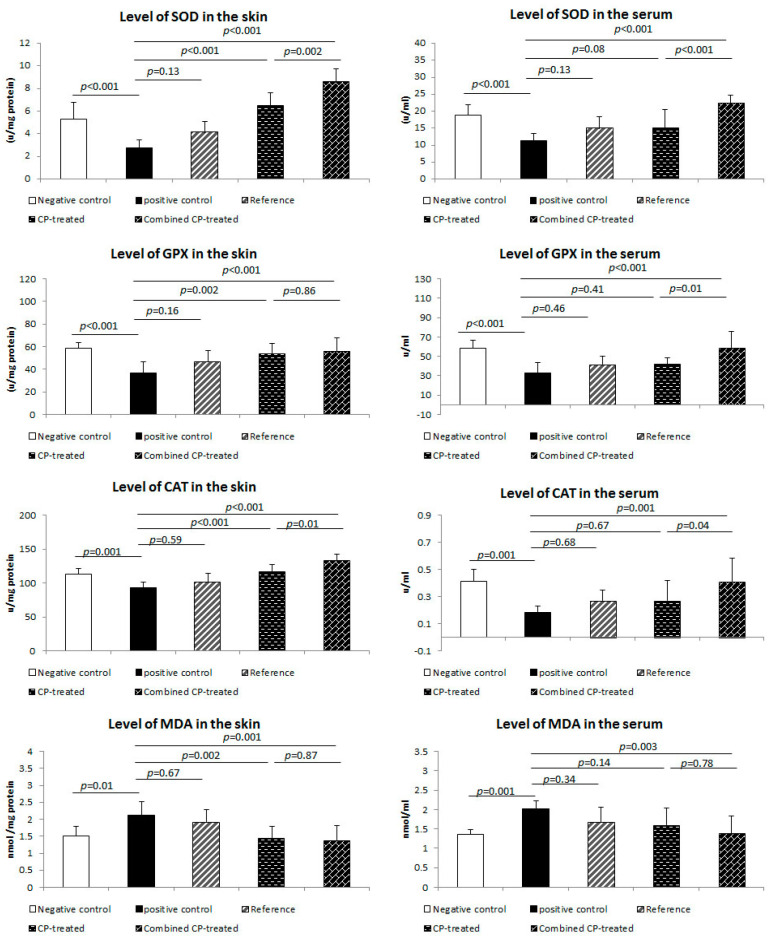
Effect of *Cucurbita pepo* L. (CP) on the oxidant/antioxidants profile of the study groups. Levels of SOD, GPX, CAT, and MDA in the skin wound were assessed using ELISA. Data are presented as the mean ± SD, *n* = 10. One way ANOVA test was used to compare between groups, followed by the Bonferroni post hoc test.

**Figure 5 nutrients-14-03336-f005:**
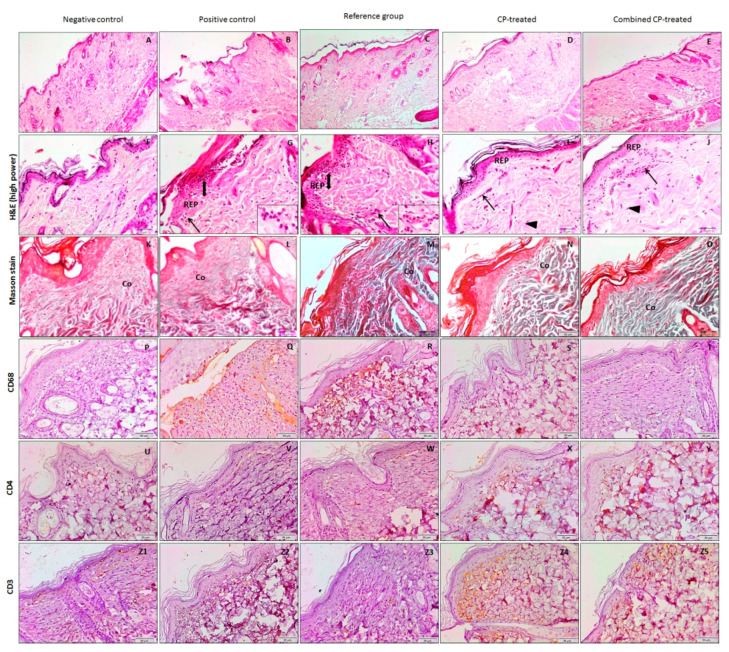
Sections of the wound area of the study groups at day 14 post wounding. The negative control group (**A**,**F**) show intact skin, while that of the positive control (**B**,**G**) and reference groups (**C**,**H**) show incomplete re-epithelization (REP), many neutrophils (bihead arrow), inflammatory cells and the insert show a higher magnification of these cells. The wound areas of CP-treated (**D**,**I**) and combined CP-treated (**E**,**J**) groups show dominated fibroblasts (arrowhead) and few inflammatory cells, and a complete newly formed epithelial layer (REP) (H&E stains X 400). Masson Stained sections show adequate, well-arranged, and compact collagen fibers (Co) in both the negative control group (**K**) and combined CP-treated groups (**L**), while the other treated groups (**M**–**O**) show irregular fewer collagen fibers (Co) (Masson stain X400). Sections of wound areas are immunohistochemically stained with CD68 (**P**–**T**), CD 4 (**U**–**Y**), and CD3 (**Z1**–**Z5**). A-EX100, F-ZdX400.

**Table 1 nutrients-14-03336-t001:** The effect of CP administration on behavioral changes, wound closure percentage, corticosterone, pro-inflammatory cytokines, and immune cell count.

Group	Negative Control Group	Positive Control Group	Reference Group	CP-Treated Group	Combined CP-Treated Group
Total immobility time (seconds)	302.11 ± 7.22	353.00 ± 30.41 ^a^	340.10 ± 20.31	331.50 ± 16.77	307 ± 22.84
Wound area (mm^2^) on day 4	00.00 ± 00.00	12.278 ± 1.38	10.75 ± 1.55 ^b^	9.90 ± 0.74 ^c^	9.50 ± 0.97
Wound area (mm^2^) on day 14	00.00 ± 00.00	6.43 ± 1.54	3.50 ± 0.85 ^b^	2.50 ± 0.71 ^c^	0.43 ± 0.03
Wound closure %	00.00 ± 00.00	47.41 ± 12.71	67.54 ± 5.99	74.25 ± 9.00 ^c^	87.00 ± 4.12
Corticosterone level in serum (ng/mL)	5.67 ± 1.25	11.28 ± 1.80 ^a^	10.81 ± 1.57	9.87 ± 2.01	6.37 ± 1.07
IL-6 in skin (pg/mg protein)	19.19 ± 4.06	82.44 ± 10.28 ^a^	65.13 ± 18.90 ^b^	49.49 ± 15.77 ^c^	30.82 ± 7.57
TNF-α in skin (pg/mg protein)	31.60 ± 6.39	98.65 ± 21.61 ^a^	72.89 ± 18.08 ^b^	55.22 ± 13.91 ^c^	35.57 ± 9.01
IL-6 serum (pg/mL)	25.97 ± 3.86	115.82 ± 14.60 ^a^	101.55 ± 15.07	101.17 ± 12.35	35.39 ± 6.45
TNF-α in serum (pg/mL)	29.58 ± 7.84	110.00 ± 13.39 ^a^	97.52 ± 9.63	97.90 ± 8.41	40.74 ± 7.91
COX-2 in skin (ng/mg protein)	0.88 ± 0.19	3.97 ± 1.27	3.12 ± 1.21	2.49 ± 1.04 ^c^	1.15 ± 0.48
iNOS in skin (u/mg protein)	0.71 ± 0.28	3.33 ± 1.36 ^a^	2.43 ± 0.90	2.17 ± 0.64 ^c^	1.05 ± 0.49
Number of CD68 positive cells (cell/mm^2^)	212.10 ± 31.89	762.80 ± 44.94 ^a^	717.00 ± 48.72	394.20 ± 23.09 ^c^	320.70 ± 59.28
Number of CD3 positive cells (cell/mm^2^)	394.70 ± 113.15	243.70 ± 58.50 ^a^	366.50 ± 86.48	394.60 ± 52.69 ^c^	539.00 ± 154.09
Number of CD4 positive cells (cell/mm^2^)	404.20 ± 44.67	217.20 ± 54.60 ^a^	310.40 ± 96.65	494.30 ± 172.16 ^c^	673.90 ± 153.78
Area percent of Masson-stained collagen fibers	10.61 ± 1.49	15.60 ± 2.95 ^a^	12.80 ± 2.86	19.60 ± 2.50 ^c^	21.90 ± 3.96

An ANOVA test was used to compare the study groups, followed by the Bonferroni post hoc test. Results are presented as mean ± standard deviation (SD). Significance was considered at *p* < 0.05. ^a^ significant-compared to the negative control group. ^b^ significant-compared to the positive control group. ^c^ significant compared to the CP-treated group.

## Data Availability

The data of this work will be made available upon request.

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
