# Peer review of "Accelerating Effect of Cucurbita pepo L. Fruit Extract on Excisional Wound Healing in Depressed Rats Is Mediated through Its Anti-Inflammatory and Antioxidant Effects"

_nutrients, 2022, doi:10.3390/nu14163336_

Round 1

Reviewer 1 Report

The manuscript reported the roles of the ethanolic extraction of Cucurbita pepo L. fruit in the improvement of excisional wound healing in depressed rats. The antioxidant, anti-inflammatory and depressive markers in serum and skin from the wound edges were analyzed to explore its mechanisms of action. These results seem to be interesting but there are some major concerns in the manuscript that need to be addressed. 

Major concerns

-        The manuscript requires extensive grammar correction.

-        Role of inflammation in depression should be described in the introduction section.

-        The authors mentioned the roles of astringent and hemostatic agents as well as fluoxetine in wound healing. However, no association of these compounds with Cucurbita pepo L. was stated.

-        Since the CP fruit extract was used, it should be stated in the title.

-        Although the extraction process was previously reported, it should be described briefly in section 2.1.

-        What is the major bioactive compound(s) in the extract that might be responsible for antidepressant, antioxidant and anti-inflammatory activities of CP as the authors stated in the discussion section? The authors proposed that phenolic and flavonoids compounds may act as antioxidant and anti-inflammation agents to improve wound healing. However, there are no reports for the amount of those compounds in CP used in the present study. The component of Cucurbita pepo L. extract was identified using GC-MS analysis which mainly focuses on fatty acids and volatile organic compounds. Therefore, the total polyphenol and total flavonoid content of CP extract must be reported.

-        Since the extract was stored in a glassy container, does it affect the moisture content and the amounts of bioactive compounds? How do the authors control the quality of the extract?

-        Why only the dosages of CP at 2% (w/w) for tropical use and at 100 mg/kg for oral administration were selected for the experiments?

-        According to the results, blood samples were analyzed for biochemical markers only at the end of the experiment. However, section 2.6 mentioned that blood samples during the experiment were also collected. Since the normal ranges of biochemical parameters measured in blood collected from heart and intra-orbital sinus are different, the authors should describe which biochemical parameter(s) that was measured during and at the end of the experiments?

-        The statistical significance presented in Table 1 seems confusing. Since p < 0.05 was considered as statistical significance, the exact p values don’t really need to be presented for all comparisons. The table should be revised by using different letters (a, b, c) to indicate statistically significant differences between groups.

-        The result description in the text is confusing and need to be improved

-        In discussion, the authors mentioned that mucilage content of pumpkin is able to accelerate wound healing. However, mucilages are water soluble and practically insoluble in ethanol which was the extraction solvent of CP in the present study. Therefore, the discussion needs to be revised.

-        Conclusion

o   Please change “this study revealed that Cucurbita pepo L. succeeded…” to “…that the ethanolic extract of Cucurbita pepo L. fruit succeeded…”.

o   The conclusion should summarize the findings of the study to support the claims as antioxidant, antiinflammatory and anti-depressant activities.

Minor concerns

·       Scientific name of the plant should not be italicized through the article.

·       Please change “Cucurbita pepo L” to “Cucurbita pepo L.”  

·       Results section in abstract:  Please change “Oral administration of CP significantly reduced…” to “The administration of CP both oral and tropical significantly reduced…”.

·       The quotation mark can be removed from the introduction and discussion section, section 2.8, 2.9,  and 2.10.

·       Section 2.6 can be revised to make it more concise.

·       Section 2.8: How the data was expressed?

·       Section 2.7 to 2.9 can be rearranged, with describing tissue dissection and preparation first followed by the analysis of biochemical changes, PCR, and histology in the skin.

·       Section 2.9: Please revise to“Anti CD3, CD4, CD68 (Biocare Medical, Pacheco, USA, at dilution of 1: 100) were used.”

·       Section 2.10: “…post hoc test to avoid a multiple-comparison effect.” ??

·       Section 3.2: Please change

o   “Figure (2)” to “Figure 2”.

o   Revise the sentence to “the values at day 4 and 14 were shown in Table 1”.

o   “The wound closure” to “The percentage wound closure”

·       Section 3.2: “It was significantly high (pË‚0.001) in all the treated groups compared to the positive control group with the highest value in the combined CP-treated group.”. This sentence is confusing and should be revised.

·       Table 1: non-capital letter for “skin”

·       Please correct “the studied groups” to “the study groups”

·       Page 9, paragraph 2: Please change “Figure 2” to “Figure 4”

·       Figure 4: Please add the y-axis label for GPX in serum

·       Section 3.6: Please change “Figure (5)” to “(Figure 5)”

·       Discussion: Full term of PMN should be provided before using the abbreviation form.

Author Response

Dear respected editor

Thank you for giving me the chance to improve my manuscript after doing the changes requested by the respectable reviewers.

Comments

Response

Major concerns

-        The manuscript requires extensive grammar correction.

Done

-        Role of inflammation in depression should be described in the introduction section.

It is already present in the introduction line 64.

“ro-inflammatory cytokines e.g. IL-1β, IL-6, TNF- α and acute phase proteins were described to be increased in experimental animal models of chronic stress-induced depression”

The authors mentioned the roles of astringent and hemostatic agents as well as fluoxetine in wound healing. However, no association of these compounds with Cucurbita pepo L. was stated.

It was reported that pumpkin works as an antiseptic, astringent and anti-inflammatory and provides detoxifying, soothing and toning support (Safar 2019).

Safar A.A. The Role of pumpkin Seed oil in Healing of Wounds in Mice. Journal of the University of Garmian 6 (1), 2019.

  Since the CP fruit extract was used, it should be stated in the title.

Added

Although the extraction process was previously reported, it should be described briefly in section 2.1.

It was added in details.

The skin and pulp of the fruit were dried by a lyophilize machine freeze-drier and crushed by a grinding electrical machine. The dried powder (50 g) was mixed with 450mL ethanol (80%) for 1 day at 37°C temperature, left in a shaker machine for another day, and then filtered twice. The extract was left at a fume hood to allow extra evaporation of ethanol; then, the extract was dried in a freeze-drier.

What is the major bioactive compound(s) in the extract that might be responsible for antidepressant, antioxidant and anti-inflammatory activities of CP as the authors stated in the discussion section? The authors proposed that phenolic and flavonoids compounds may act as antioxidant and anti-inflammation agents to improve wound healing. However, there are no reports for the amount of those compounds in CP used in the present study. The component of Cucurbita pepo L. extract was identified using GC-MS analysis which mainly focuses on fatty acids and volatile organic compounds. Therefore, the total polyphenol and total flavonoid content of CP extract must be reported.

Yes this is true.

 We did identify the component of Cucurbita pepo L. extract was using GC-MS analysis which mainly focuses on fatty acids and volatile organic compounds. It was reported oleic and linoleic acids are among the compounds that exhibit strong pro-inflammatory activities and play a major role in the recruitment of inflammatory cells at the inflammation site that lead to accelerate the wound healing process.

Regarding the phenolic and flavonoids compounds we did not analyze them but we depend on and cited the references that document its presence in the pumpkin.

Since the extract was stored in a glassy container, does it affect the moisture content and the amounts of bioactive compounds? How do the authors control the quality of the extract?

No, the moisture content and the amounts of bioactive compounds were not affected because the extract was dried in a freeze-drier machine (FD5508; ILShinBase Co., Ltd., Korea).

This was added in the detailed process of extraction.

 Why only the dosages of CP at 2% (w/w) for tropical use and at 100 mg/kg for oral administration were selected for the experiments?

Because these concentrations proved an efficacy in previous studies, therefore we cited them.

Bardaa, S., et al., The evaluation of the healing proprieties of pumpkin and linseed oils on deep second-degree burns in rats. Pharmaceutical biology, 2016. 54(4): p. 581-587.

Balgoon, M.J., et al., Combined Oral and Topical Application of Pumpkin (Cucurbita pepo L.) Alleviates Contact Dermatitis Associated With Depression Through Downregulation Pro-Inflammatory Cytokines. Frontiers in Pharmacology, 2021. 12(898).

According to the results, blood samples were analyzed for biochemical markers only at the end of the experiment. However, section 2.6 mentioned that blood samples during the experiment were also collected. Since the normal ranges of biochemical parameters measured in blood collected from heart and intra-orbital sinus are different, the authors should describe which biochemical parameter(s) that was measured during and at the end of the experiments?

It was specified as follow.

Serum corticosterone level was assessed, during the experiment in order to confirm the occurrence of depressive like status,

The statistical significance presented in Table 1 seems confusing. Since p < 0.05 was considered as statistical significance, the exact p values don’t really need to be presented for all comparisons. The table should be revised by using different letters (a, b, c) to indicate statistically significant differences between groups.

Done

a: significance versus the negative control group

b: significance versus positive control group

c : significance versus CP-treated group

-        The result description in the text is confusing and need to be improved

Done

In discussion, the authors mentioned that mucilage content of pumpkin is able to accelerate wound healing. However, mucilages are water soluble and practically insoluble in ethanol which was the extraction solvent of CP in the present study. Therefore, the discussion needs to be revised.

This is possible from the practical point of view. It is observed in my study as well as many of the previous studies e.g. Bahramsoltani et al., 2017.

It was found that  in a mechanistic point of view, the wound-healing activity of C. moschata peel extract could be attributed to its high mucilage content and presence of different constituents such

as flavonoids and phenolic compounds that are able to accelerate wound healing as well as its antioxidant power (Bahramsoltani et al., 2017).

Bahramsoltani R, Farzaei MH, Abdolghaffari AH, Rahimi R, Samadi N, Heidari M, Esfandyari M, Baeeri M, Hassanzadeh G, Abdollahi M, Soltani S, Pourvaziri A, Amin G. Evaluation of phytochemicals, antioxidant and burn wound healing activities of Cucurbita moschata Duchesne fruit peel. Iran J Basic Med Sci. 2017

Conclusion

o   Please change “this study revealed that Cucurbita pepo L. succeeded…” to “…that the ethanolic extract of Cucurbita pepo L. fruit succeeded…”.

Done

o   The conclusion should summarize the findings of the study to support the claims as antioxidant, antiinflammatory and anti-depressant activities.

It is already present in line 494

Minor concerns

·       Scientific name of the plant should not be italicized through the article.

Done

Please change “Cucurbita pepo L” to “Cucurbita pepo L.”  

Done

Results section in abstract:  Please change “Oral administration of CP significantly reduced…” to “The administration of CP both oral and tropical significantly reduced…”.

Done

The quotation mark can be removed from the introduction and discussion section, section 2.8, 2.9,  and 2.10.

Done

Section 2.6 can be revised to make it more concise.

Done

·       Section 2.8: How the data was expressed?

Added

The level of mRNA was presented as a

ratio or percent to that of corresponding GAPDH

Section 2.7 to 2.9 can be rearranged, with describing tissue dissection and preparation first followed by the analysis of biochemical changes, PCR, and histology in the skin.

I don’t understand what is meant by that. The presentation is already in this order mention.

Section 2.9: Please revise to“Anti CD3, CD4, CD68 (Biocare Medical, Pacheco, USA, at dilution of 1: 100) were used.”

Done

Section 2.10: “…post hoc test to avoid a multiple-comparison effect.” ??

Removed

Section 3.2: Please change

o   “Figure (2)” to “Figure 2”.

o   Revise the sentence to “the values at day 4 and 14 were shown in Table 1”.

o   “The wound closure” to “The percentage wound closure”

Done

Section 3.2: “It was significantly high (pË‚0.001) in all the treated groups compared to the positive control group with the highest value in the combined CP-treated group.”. This sentence is confusing and should be revised.

Revised and corrected.

·       Table 1: non-capital letter for “skin”

Done

Please correct “the studied groups” to “the study groups”

Done

·       Page 9, paragraph 2: Please change “Figure 2” to “Figure 4”

Done

·       Figure 4: Please add the y-axis label for GPX in serum

Done

Section 3.6: Please change “Figure (5)” to “(Figure 5)”

Done

Discussion: Full term of PMN should be provided before using the abbreviation form.

Done

Reviewer 2 Report

Your work deserves credit in spite experimental data can lack a lot of soundness for evidence building. The experimental design is adequate and your methods had covered all the possibilities in terms of analysis outcomes. 

In spite of this, the manuscript needs some significant improvements: 

- Please add line number for further revisions

Introduction 

you say depression is a major illness that can be fatal, in fact it can increase suicide risk but depression itself is not been proven to be fatal. 

Nevertheless, I would recommend you focus more on benefits from polyphenols in wound healing (including infection control) and not depression, anxiety or other diseases considering that it requires further explanations on how is it related. 

The paragraph "Although there are many topical preparations ...." end with the word "are" that is a verb, probably it is lacking some text. 

In the last paragraphs you refer to pumpkin for the first time, it would be important to first refer it as the species you are using. 

Materials and methods

I would suggest you start with the global experimental design description: species, number, days , etc. and ethical approval. Then how you did the random distribution and finally the group distribution 

For the sentence "Extraction of CP was done according to the method of [20] ", it would be better to say "according with the method described by Wang et al. (20)". 

The same in 2.5. 

Results

Avoid using "claims" as sub section titles

Try to put Table 1 in just one page or divide it but keep the headers in the 2nd page where it continues. 

Before 3.5. it is "These findings" probably 

I could not find Figure 5 referred in any part of results. 

Discussion

Start this section with a brief paragraph on the aim and main results from your work. 

Was this test protocol you used a gold standard to induce CUMS? Can you justify it? 

Which compounds can be present in this species that can be attributed these effects? Can you find similar studies using those compounds? 

Don't you think water immersion as used in this text could interfere with the healing? 

Additionally you have some different formatting within in the text like different color highlight in numbers. 

I highlighted some of this corrections in the pdf version, I hope it helps. 

Author Response

Dear respected editor

Thank you for giving me the chance to improve my manuscript after doing the changes requested by the respectable reviewers.

Comments

Response

Please add line number for further revisions

Added

Introduction

you say depression is a major illness that can be fatal, in fact it can increase suicide risk but depression itself is not been proven to be fatal.

That is true.

Thank you for this valuable comment. It is fixed

Nevertheless, I would recommend you focus more on benefits from polyphenols in wound healing (including infection control) and not depression, anxiety or other diseases considering that it requires further explanations on how is it related.

Regarding the advice to focus more on the infection control benefits of pumpkin in wound healing, it was added.

“it was reported that The different types of pumpkin leaves extracts, water, ethanol, hexane, showed a potent antibacterial activity as they inhibit the growth of many bacteria, e.g. Escherichia coli, Klebsiella pneumonia, Staphylococcus aureus, Proteus mirabilis and Pseudomonas aeruginosa as well as fungi e.g. Aspergillus fumigatus, Aspergillus niger, and Candida albicans comparable with the known antibiotic Ciprofloxacin and the antifungal drug Kenazole (Mohammed et al., 2018).

Regarding the explanations on how depression, anxiety or other diseases is related to polyphenol in general and Cucurbita pepo L. in specific. The relation is the anti-inflammatory activity of these compounds explains its ability to modulate the diseases that was having inflammatory nature in its pathogenesis e.g. depression, anxiety.

“Pro-inflammatory cytokines e.g. IL-1β, IL-6, TNF- α and acute phase proteins were reported to be increased in experimental animal models of chronic stress-induced depression [5]”

[10] Mohammed, H., Najem, R. S. and Altekrity, S. S. A.. Antimicrobial and antifungal activity of pumpkin (Cucurbita pepo) leaves extracted by four organic solvents and water. Iraqi Journal of Veterinary Sciences, 2018. 32: p. 33-39

The paragraph "Although there are many topical preparations ...." end with the word "are" that is a verb, probably it is lacking some text.

Corrected

In the last paragraphs you refer to pumpkin for the first time, it would be important to first refer it as the species you are using.

Done

Materials and methods

I would suggest you start with the global experimental design description: species, number, days , etc. and ethical approval. Then how you did the random distribution and finally the group distribution

Done

For the sentence "Extraction of CP was done according to the method of [20] ", it would be better to say "according with the method described by Wang et al. (20)".

Done

The same in 2.5.

Done

Results

Avoid using "claims" as sub section titles

Done

Try to put Table 1 in just one page or divide it but keep the headers in the 2nd page where it continues.

Done

Before 3.5. it is "These findings" probably

Done

I could not find Figure 5 referred in any part of results.

Its referred in line 342. Masson stain revealed the presence of adequate, well arranged and compact collagen fibers Figure (5).

Discussion

Start this section with a brief paragraph on the aim and main results from your work.

It is already present starting from line 379.

“This study was performed to assess the role of the combined local and systemic administration of Cucurbita pepo in enhancing wound healing in animal model of depression induced by exposure to chronic stress and explore the mechanism behind this role.

In the current work, rats after CUMS exposure exhibited depressive-like status evidenced by increased immobility durations during the FST as well as the significant increase in serum corticosterone”

Was this test protocol you used a gold standard to induce CUMS? Can you justify it?

The CUMS model was used in this study as it has become firmly established as an indispensable experimental tool for studying the neurobiological basis of depression with proved validity and reliability. (Willner 2016) [33].

[34] Willner P. The chronic mild stress (CMS) model of depression: History, evaluation and usage. Neurobiol Stress. 2016 Aug 24;6:78-93. doi: 10.1016/j.ynstr.2016.08.002. PMID: 28229111; PMCID: PMC5314424.

Which compounds can be present in this species that can be attributed these effects? Can you find similar studies using those compounds?

Done. It was added

Oleic and linoleic acids are among the compounds that exhibit strong pro-inflammatory activities and play a major role in the recruitment of inflammatory cells at the inflammation site that lead to accelerate the wound healing process. Fortunately, pumpkin possess high concentration of these two compounds (Quirino et al., 2009, Pereira et al., 2008).

51. Pereira LM, Hatanaka E, Martins EF, et al. (2008). Effect of oleic and linoleic acids on the inflammatory phase of wound healing in rats. J Cell Biochem Funct 26:197–204.

52. Quirino JS, Leite GO, Rebelo LM, et al. (2009). Healing potential of Pequi (Caryocar coriaceum Wittm.) fruit pulp oil. J Phytochem Lett 92:1–5.

Don't you think water immersion as used in this text could interfere with the healing?

I don’t think so. As water immersion was for 6 minutes only which is a very short time mimic hand washing.

Additionally you have some different formatting within in the text like different color highlight in numbers.

It is standardized now.

I highlighted some of this corrections in the pdf version, I hope it helps.

It was done. Thank you
